

# Reactive Oxygen Species Build-up in Photochemically Aged Iron-and Copper-doped Secondary Organic Aerosol Proxy

Kevin Kilchhofer[1, 2], Alexandre Barth[3], Battist Utinger[3], Markus Kalberer[3], and Markus Ammann[1]

[1]PSI Center for Energy and Environmental Sciences, 5232 PSI Villigen, Switzerland
[2]Department of Environmental System Science, Institute for Atmospheric and Climate Science, ETH Zurich, 8092 Zurich, Switzerland
[3]Department of Environmental Sciences, University of Basel, 4056 Basel, Switzerland

**Correspondence:** Kevin Kilchhofer (kevin.kilchhofer@psi.ch)

**Abstract.** The toxicity of particulate matter (PM) is highly related to the concentration of particle-bound reactive oxygen species (ROS). Chemical properties, including metal dissolution and the sources of PM, influence ROS production and its oxidative potential. Here, the photochemical aging of a secondary organic aerosol proxy (citric acid, CA) with metal complexes (iron-citrate, $Fe^{III}Cit$) is assessed toward the production of particle-bound ROS with an online instrument (OPROSI). We

5  studied the photochemically induced redox chemistry in iron/copper-citrate particles experimentally mimicked with an aerosol flow tube (AFT) in which UV-aging was probed. Different atmospheric conditions were tested, influencing the physicochemical properties of the particles. We found that UV-aged CA aerosol containing 10 mole % $Fe^{III}Cit$ generated ROS concentrations on the order of $0.1 \, \mathrm{nmol} \, H_2O_2 \, \mathrm{eq} \, \mu \mathrm{g}^{-1}$, indicating the photochemically driven formation of peroxides. An increase in relative humidity (RH) leads to only a slight but overall lower concentration of ROS, possibly due to a loss of volatile $HO_2$ and $H_2O_2$

10  in the gas phase in the less viscous particles. The RH effect is enhanced in nitrogen sheath flow, but in air and compared to the $Fe^{III}Cit$/CA particles, the iron/copper-citrate samples show a uniformly decreased ROS level. Interestingly, in the high humid nitrogen experiment with copper, we found a much more pronounced decline of the ROS concentration down to $2 \times 10^{-2} \, \mathrm{nmol} \, H_2O_2 \, \mathrm{eq} \, \mu \mathrm{g}^{-1}$ compared to all other irradiation experiments. We suggest that copper may suppress radical redox reactions and therefore consume ROS in an anoxic regime.





## 1 Introduction

Atmospheric particulate matter (PM) is highly associated with adverse health effects that cause respiratory disease, cardiovascular disease, and cancer (Dockery and Pope, 1994; Laden et al., 2006; Lepeule et al., 2012). Urban areas around the world (especially India and China) are particularly affected by such adverse health effects induced by oxidative stress (Lelieveld et al., 2020). However, our understanding of the physical and chemical properties of PM that lead to oxidative stress upon exposure remains incomplete (Bates et al., 2019). Oxidative stress is defined by an imbalance between increased levels of reactive oxygen species (ROS) and a low activity of antioxidant mechanisms (Preiser, 2012; Donaldson et al., 2001; Li et al., 2003). Since long, it has been known that PM can produce ROS (Oettinger et al., 1999), and the origins of ROS have been associated with direct production by the particles themselves or by PM activated leukocytes (Prahalad et al., 1999). ROS are any oxygen-containing molecules that have one or more unpaired electrons, making them highly reactive (including OH, $HO_2$ and $H_2O_2$ species), and key drivers of oxidative stress (Knaapen et al., 2004). These reactive species can be introduced into the body by inhaling PM that contains ROS (particle-bound, exogeneous ROS) or, as discovered by Dellinger et al. (2001), can be generated internally through a catalytic process after inhaling redox-active PM species (endogeneous ROS). This process was defined as oxidative potential (OP) by Bates et al. (2019). Thus, OP, in comparison to the mass concentration with particle size and composition, has been suggested to be a more health-relevant metric (e.g. Yang et al. (2015); Yadav and Phuleria (2020)).

The sources and composition of PM that produce ROS and OP are extensively studied. Daellenbach et al. (2020) summarized that OP in Europe are mostly associated with anthropogenic emissions such as secondary organic aerosols (SOA) largely from residential biomass burning and coarse-mode metals from vehicular non-exhasut emissions. Recently, OP assessments were also done for total outdoor $PM_{2.5}$ in Fairbanks, Alaska (Yang et al., 2024) as the impact of non-anthropogenic PM was recently highlighted by the World Health Organization (Pai et al., 2022). Tuet et al. (2019) found open biomass burning in the Brazilian Amazon cause high levels of ROS concentrations and thus, oxidative stress. Furthermore, for instance, photochemically aged organic aerosol (OA) throughout atmospheric transport showed substantially different OP than non-aged samples collected during fires in Greece, with both increasing and decreasing effect (Wong et al., 2019). This means that the relationship between $PM_{2.5}$ mass and OP is largely non-linear. Salana et al. (2024) determined that this phenomenon occurs because of notable variations in intrinsic toxicity, which stem from the spatially heterogeneous chemical composition of the aerosol.

Only a few studies, however, have probed the chemical interactions of particle-bound ROS with redox-active transition metals (Charrier et al., 2014; Gonzalez et al., 2017; Wang et al., 2018), even though soluble metals were suggested to be strongly linked to the OP of aerosols (Fang et al., 2017; Lelieveld et al., 2021; Tong et al., 2021; Tacu et al., 2021; Campbell et al., 2023). Wei et al. (2019) found that processes such as complex formation with organic ligands influences metal solubility and thus, redox chemistry. Indirect measurements and model results reported ROS build-up of a metal complexed SOA proxy during photochemical aging processes (Dou et al., 2021; Alpert et al., 2021; Kilchhofer et al., 2024) and heterogeneous photochemistry contributes to the oxidant budget in atmospheric particles and thus leads to the formation of particle-bound ROS (Corral Arroyo et al., 2018). A review by Al-Abadleh (2024) re-emphasized the significance of iron dissolved in ambient OA particles. Natural emissions from dust regions and anthropogenic activities such as traffic and combustion processes are the main sources of





soluble iron (Ito and Miyakawa, 2023). Also, copper emissions increased greatly during the industrial revolution (Hong et al.,

50     1996), and the atmospheric copper concentration was quantified as up to a tenth of the ambient iron concentration (Schroeder et al., 1987). Understanding the distinct functions of PM components and the chemical processes they initiate is crucial towards a fundamental understanding of ROS formation (Shiraiwa et al., 2017). Here, we focus on examining the formation of particle-bound ROS within an iron and/or copper containing SOA proxy in a non-and UV-aged scenario under changing atmospheric conditions.

55     There are a range of acellular assays that are utilized to measure ROS (Fuller et al., 2014). We chose an automated on-line particle-bound ROS instrument (OPROSI) developed by Wragg et al. (2016), which uses the commonly applied 2'7'-dichlorofluorescein (DCFH) with horseradish peroxidase (HRP) as acellular assay (Calas et al., 2018; Bates et al., 2019). DCFH is sensitive to $H_2O_2$ and organic peroxides (Fuller et al., 2014), but not to redox-active transition metals like iron and copper. A key reason for employing an online measurement device was the capability to measure ROS concentrations with high

60     temporal resolution, enabling the tracking of rapidly changing atmospheric conditions such as humidity and UV irradiation. Furthermore, it is shown that up to 90% of particle-bound ROS are lost prior to offline analysis, after collection on filter and extraction (Zhang et al., 2022; Campbell et al., 2023).



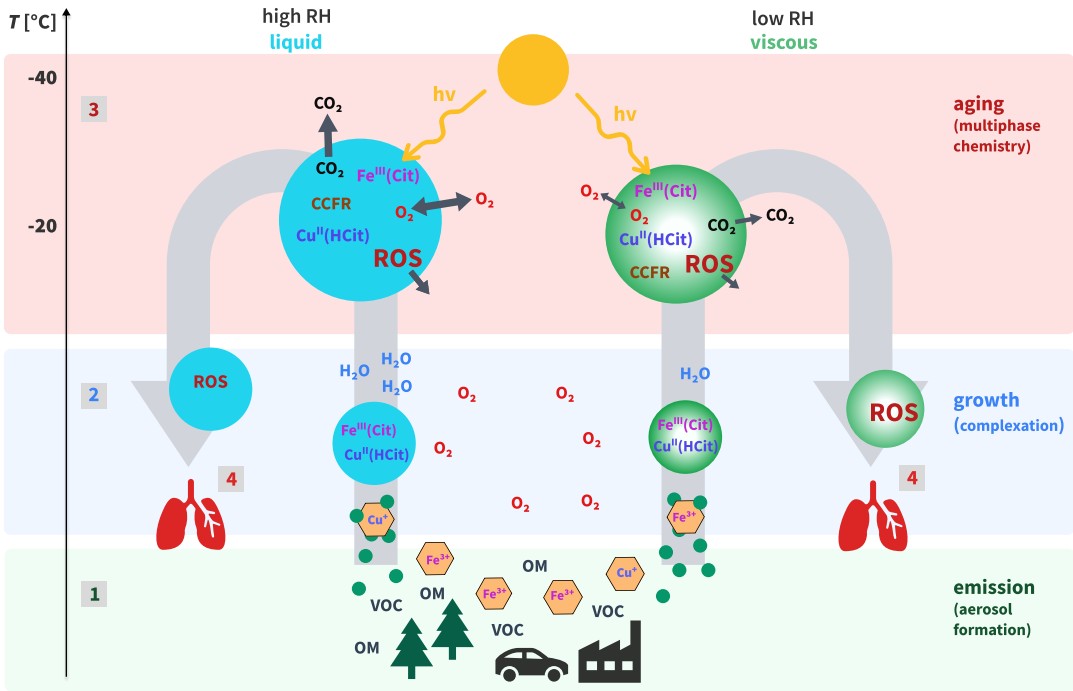

**Figure 1.** Schematic of the photochemical aging process of an iron(III)-citrate/copper(II)-citrate/ citric acid (Fe$^{III}$Cit/Cu$^{III}$HCit/CA) particle with the induced multiphase chemistry. The left pathway exemplary depict an OA growth in high RH conditions implying a liquid OA phase, whereas on the right the particles experience low RH conditions that leads to highly viscous organic phase. The numbers 1-4 reference the steps the particles undergo during their atmospheric lifetime and correspond to the experimental steps in the laboratory setup shown in Figure 2. Number 4 implicates the possible ROS build-up during the UV aging in either liquid or highly viscous aerosols inducing oxidative stress in human lungs. OM: organic material; VOC: volatile organic carbon; CCFR: carbon-centered free radical; ROS: reactive oxygen species (see Table 1 for details).

We report online quantifications of particle-bound ROS concentrations in photochemically aged citric acid (CA) particles doped with iron and / or copper. For this purpose, we experimentally mimicked the processes experienced by an aerosol during its atmospheric lifetime, as shown in Figure 1. The processes are divided into aerosol formation including the dissolution of transition metals (1), aerosol growth at certain relative humidity (2), photochemistry triggered by UV irradiation (3) and ROS accumulation (4). The phochemical mechanism of Fe$^{III}$Cit is simplified in Table 1. Under light the Fe$^{III}$Cit complex is excited into a reduced Fe(II) radical complex, which may decay into Fe$^{2+}$ ions and a citrate radical. The citrate radical rapidly decays by decarboxylation of $CO_2$ yielding a carbon-centered free radical (CCFR, $^\bullet C_5H_5O_5^{2-}$) (R1). Oxygen adds to the CCFR to form a short-lived peroxyradical that leads to oxidation of the alcohol group to a ketone ($C_5H_4O_5^{2-}$) and superoxide (ROS) in reaction 2. Reactions 3-8 describe ROS cycling and 9-12 Fe$^{II}$ oxidant reactions. As schematically shown in Figure 1, we expected lower particle-bound ROS concentrations in aerosols photochemically aged under humid conditions as diffusion processes enhance gas-particle phase exchange and thus loss to the gas phase. The results show very elevated



ROS concentrations in photochemically aged iron(III)-citrate (Fe$^{III}$Cit) particles compared to pure CA or non-aged particles. However, humidity and presumably molecular diffusion did not influence ROS formation as much as previously hypothesized and reported. UV-aged copper-containing Fe$^{III}$Cit particles disclosed an unexpectedly high change in ROS formation depending on oxygen availability. The limitation of using DCFH as an acellular assay was tried to overcome with additional experiments using the online oxidative potential ascorbic acid instrument (OOPAAI, Utinger et al. 2023). However, AA directly interacted with copper-inducing OP and thus was not suitable for quantifying particle-bound ROS. The high oxidation capacity of metals is stressed, as is the influence of metal interactions during atmospheric aging processes toward the formation of ROS in particles.

**Table 1.** Mechanism of initial Fe$^{III}$Cit photochemistry. In R2, $^{\bullet}O_2^-$ and $H^+$ forms $HO_2^{\bullet}$ available in the ROS and Fe(II) oxidant reactions. $^{\bullet}O_2^-$, $HO_2^{\bullet}$, $HO^{\bullet}$, $H_2O_2$ = reactive oxygen species (ROS); $^{\bullet}C_5H_5O_5^{2-}$ = carbon-centered free radical (CCFR); $C_5H_4O_5^{2-}$ = ketone.

| Number | Reactions | References |
|---|---|---|
| R1 | $FeC_6H_5O_7 \xrightarrow{h\nu} Fe^{2+} + {}^{\bullet}C_5H_5O_5^{2-} + CO_2$ | Dou et al. (2021) |
| R2 | ${}^{\bullet}C_5H_5O_5^{2-} + O_2 \longrightarrow C_5H_4O_5^{2-} + {}^{\bullet}O_2^- + H^+$ | Hug et al. (2001) |
| | **ROS Reactions:** | |
| R3 | $HO_2^{\bullet} + HO_2^{\bullet} \longrightarrow H_2O_2 + O_2$ | Bielski et al. (1985) |
| R4 | $HO^{\bullet} + HO^{\bullet} \longrightarrow H_2O_2$ | Sehested et al. (1968) |
| R5 | $HO_2^{\bullet} + HO^{\bullet} \longrightarrow H_2O + O_2$ | Sehested et al. (1968) |
| R6 | $HO^{\bullet} + O_2^{\bullet-} \longrightarrow HO^- + O_2$ | Sehested et al. (1968) |
| R7 | $H_2O_2 + O_2^{\bullet-} \longrightarrow HO_2^{\bullet} + H_2O$ | Christensen et al. (1982) |
| R8 | $HO_2^{\bullet} + O_2^{\bullet-} \xrightarrow{H^+} H_2O_2 + O_2$ | Bielski et al. (1985) |
| | **Fe(II) Oxidant Reactions:** | |
| R9 | $Fe^{2+} + O_2^{\bullet-} \xrightarrow{2H^+} Fe^{3+} + H_2O_2$ | Rush and Bielski (1985) |
| R10 | $Fe^{2+} + HO_2^{\bullet} \xrightarrow{H^+} Fe^{3+} + H_2O_2$ | Jayson et al. (1973) |
| R11 | $Fe^{2+} + HO^{\bullet} \longrightarrow FeOH^{2+}$ | Christensen and Sehested (1981) |
| R12 | $Fe^{2+} + H_2O_2 \longrightarrow Fe^{3+} + HO^{\bullet} + HO^-$ | Walling (1975) |



## 2 Material and Methods

### 2.1 Online Particle-bound ROS Instrument

Online particle-bound ROS measurements were performed using the portable Online Particle-bound ROS Instrument (OPROSI)
as described by Wragg et al. (2016). The instrument uses the 2',7'-dichlorofluorescein (DCFH) assay to quantify the total
amount of ROS. In short, OPROSI was continuously sampling at a flow rate of $5\,\mathrm{L\,min^{-1}}$ while extracting the water-soluble
fraction of PM with $1\,\mathrm{ml\,min^{-1}}$ of a $10\,\mu\mathrm{M}$ horseradish peroxidase (HRP) solution in a 10% phosphate buffer solution (PBS).
HRP reacts immediately with ROS and is oxidized itself. This is mainly caused by $H_2O_2$ and organic hydroperoxides (Berglund
et al., 2002; Cathcart et al., 1983). This instantaneous capture eliminates any sample reactivity loss that may be an issue in
offline methods. After extraction, $1\,\mathrm{ml\,min^{-1}}$ of $10\,\mu\mathrm{M}$ DCFH in 10% PBS is added to the sample flow. The DCFH is con-
verted to DCF by the oxidized HRP. This reaction is promoted by passing the total flow through a heated bath set to $37\,^\circ\mathrm{C}$ for
a residence time, $t_r$ of $10\,\mathrm{min}$. Subsequently, the detection cell measures the fluorescence intensity of the DCF produced by
excitation of the sample at $470\,\mathrm{nm}$ and recording the emission at $520\,\mathrm{nm}$ using a spectrometer. The total amount of ROS is
given in $H_2O_2$ equivalents by calibrating the instrument's response against known concentrations of $H_2O_2$.

### 2.2 Aerosol flowtube

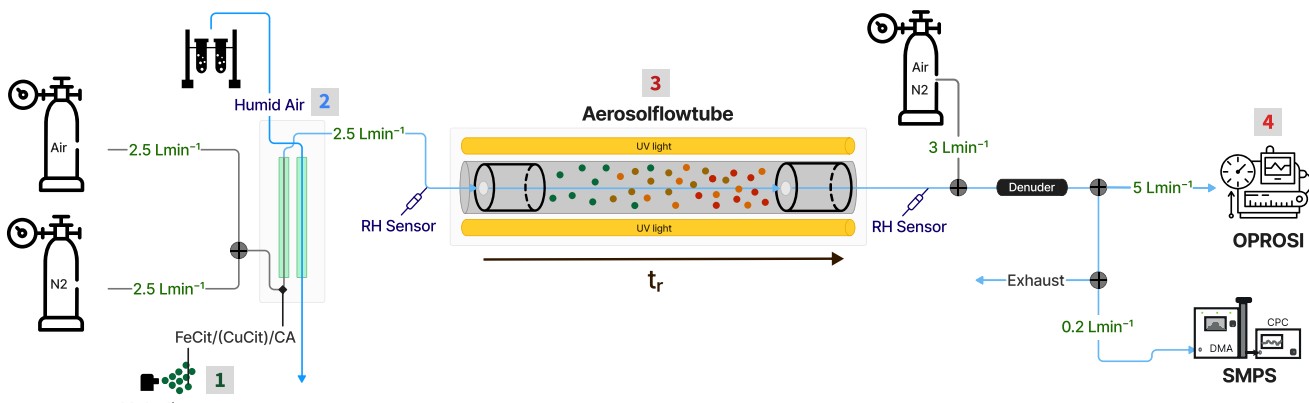

**Figure 2.** Schematic of experimental setup mimicking the photochemical aging of $Fe^{III}Cit/Cu^{III}HCit/CA$ particles. The numbers 1-4, refer-
encing the different experimental steps. The gas flow rates are indicated in green. The charcoal denuder downstream of the aerosol flowtube
(AFT) was used to eliminate the gas phase products. OPROSI = online particle-bound reactive oxygen species instrument (Wragg et al.,
2016). 1: Particle generation; 2: Particle humidification and growth; 3: Photochemical aging; 4: ROS detection. SMPS = scanning mobility
particle sizer (contains DMA = differential mobility analyzer with a CPC = condensation particle counter). The residence time ($t_r$) of the
aerosol samples in the AFT was $\approx 150\,\mathrm{s}$ and an AFT bypass line was installed for normalization.

Figure 2 schematically illustrates the experimental setup used to simulate the atmospheric process depicted in Figure 1. Aerosol
samples, as detailed in Table 2, were produced by nebulizing their precursor solution using a home-built nebulizer equipped

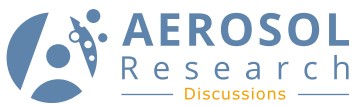

with a sonicator. Following its generation, the aerosol was brought to equilibrium at the relative humidity (RH) of the experiment using a humidified counterflow system, in which water vapor was exchanged across a permeable membrane to obtain

equilibrium between the two gas flows. The flow of the carrier gas was in $2.5\,\mathrm{L\,min^{-1}}$ of $\mathrm{N_2}$ (99.999%) or pressurized air. This gas rate was chosen to achieve a sufficiently long aerosol residence time ($t_r$) in the aerosol flow tube (AFT). The AFT consisted of a perfluoroalkoxycopolymer (PFA) tube of $7\,\mathrm{cm}$ inner diameter. It features movable teflon inserts, acting as inlet and outlet, inserted from both ends into the AFT. These symmetric inserts are conically shaped to ensure a laminar flow profile. The inlets are equipped with valves that are used for aerosol gas flow and to bypass AFT. In this work, we used a fixed length

that gives an AFT volume of about $6\,\mathrm{L}$, resulting in $t_r = 150\,\mathrm{s}$. Seven UV lamps (UVA Phillips) were surrounding the aerosol flow tube to mimic atmospheric relevant UV-aging processes (see Section 2.3). Two RH sensors measured the humidity of the gas up/-and downstream of the aerosolflowtube. A dry dilution flow of $3\,\mathrm{L\,min^{-1}}$ was added to provide sufficient sample flow for the instruments. The charcoal denuder downstream of the aerosolflowtube was used to eliminate the gas phase products. Aerosol samples were drawn into the OPROSI instrument at $5\,\mathrm{L\,min^{-1}}$ through the aerosol-conditioning unit. A scanning

mobility particle sizer (SMPS, TSI) consisting of an electrostatic classifier (Model 3082) with a differential mobility analyzer (DMA, Model 3081A) and a condensation particle counter (CPC, Model 3750) was used to measure the aerosol concentration and size distribution throughout all experiments. The SMPS inlet flow was set to $0.2\,\mathrm{L\,min^{-1}}$ and it recorded scans every $2\,\mathrm{min}$ and $45\,\mathrm{s}$. The data collected by the OPROSI, which performed scans every $20\,\mathrm{s}$, was normalized using the aerosol mass concentration. The excess flow of $0.2 \pm 0.1\,\mathrm{L\,min^{-1}}$ ensured that the AFT remained slightly above ambient pressure throughout

the experiments.

## 2.3 Actinic flux of UV lamps

The irradiance of the seven UV lamps ($I_{\mathrm{UV}}$, $\mathrm{Wm^{-2}\,nm^{-1}}$) in the AFT was measured with a UV-VIS spectrometer (AVANTES, AvaSpec-ULS2048XL-EVO) inside the center, on the left and right side of the flowtube. The irradiance data were converted into number of photons ($N_{\mathrm{photons}}$) with the photon energy ($E_{\mathrm{p}}$) as a function of wavlength $\lambda$ to obtain a photon flux density,

$E_{\mathrm{QF}}$, in $\mathrm{cm^{-2}s^{-1}nm^{-1}}$ with:

$$N_{\mathrm{photons}} = \frac{I_{\mathrm{UV}}}{E_{\mathrm{p}}(\lambda)} \tag{1}$$

$$E_{\mathrm{QF}} = \frac{N_{\mathrm{photons}}}{N_{\mathrm{A}} \cdot 1 \times 10^{-6}}, \qquad\qquad N_{\mathrm{A}} = 6.023 \times 10^{23}\,\mathrm{mol^{-1}} \tag{2}$$




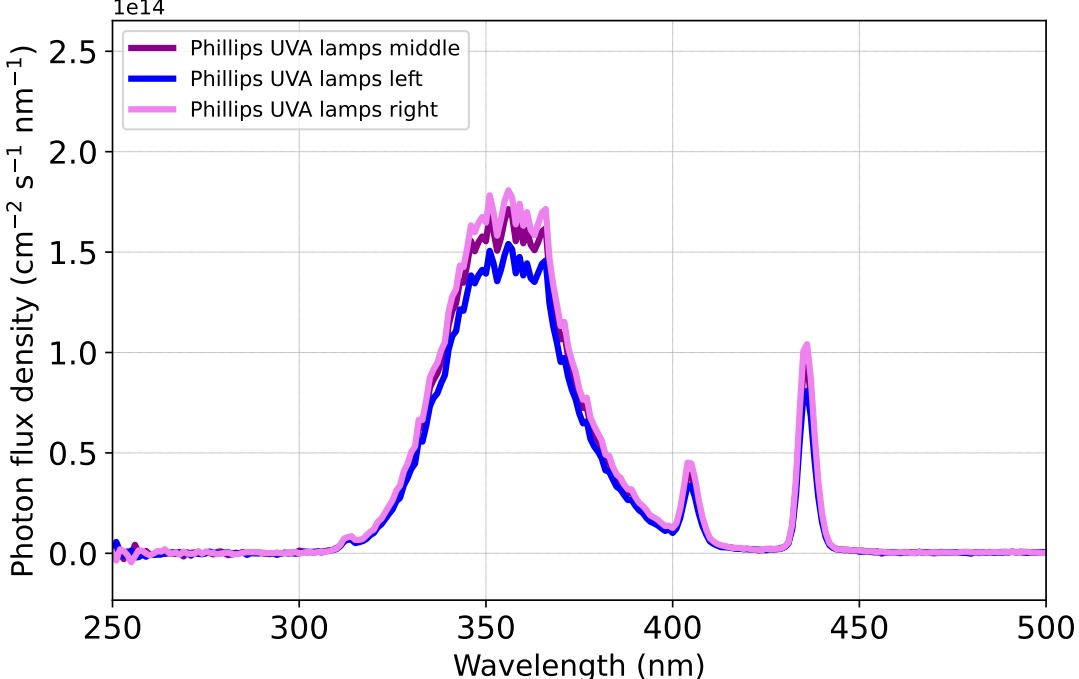

**Figure 3.** Photon flux density ($E_{QF}$, magenta), the daily average actinic flux in Los Angeles in June (orange), and the absorption cross section of the Fe$^{III}$Cit molecule (green). The wavelenght band was chosen from 300-450 nm, which is the overlap of the sun actinic flux and the UV LEDs. The y-axis on the right correspond to the cross section.

The frequency was calculated with Equation 3:

$$j_{\text{FeCit}} = \int_{\lambda_1}^{\lambda_2} \sigma_{\text{FeCit}}(\lambda) \cdot \phi_{\text{FeCit}}(\lambda, T) \cdot E_{\text{QF}}(\lambda) \, d\lambda, \tag{3}$$

where $\sigma_{\text{FeCit}}$ is the absorption cross section of a Fe$^{III}$Cit molecule (cm$^2$ molecule $^{-1}$), $\phi_{\text{FeCit}}$ the quantum yield for photolysis and $E_{QF}$ (cm$^{-2}$ s$^{-1}$ nm$^{-1}$) corresponds to the photon flux density (see Figure 3 and Equation 2). The integration was derived from $\lambda_1 = 300$ nm to $\lambda_2 = 400$ nm. This produced a photolysis frequency of $j_{\text{FeCit}} = 2.36 \pm 0.15 \times 10^{-2}$ s$^{-1}$. This frequency is almost equal to the one calculated for Los Angeles conditions at noon ($j_{\text{LA}} = 2.9 \pm 0.2 \times 10^{-2}$ s$^{-1}$).

## 2.4 Sample preparation

Citric acid (CA, $\geq$ 99.5%; CAS = 5949-29-1), Fe$^{III}$Cit tribasic monohydrate (18-20% Fe basis; CAS = 2338-05-8) and Cu$^{II}$HCit (97%; CAS = 866-82-0) were purchased from Sigma-Aldrich. The dilute aqueous solutions were prepared in ultrapure water (18 MΩcm$^{-1}$, Milli-Q). We used a pure CA solution ($1 \times 10^{-3}$ M) to establish an aerosol mass concentration needed for the



OPROSI (100-200 $\mu$g). The concentration can be fine-adjusted with the settings of the ultrasonic nebulizer (home-built). CA, Fe$^{III}$Cit and Cu$^{II}$HCit stock solutions were prepared to achieve different mole ratios (M$_r$) used in different experiments, as

listed in Table 2. The light-sensitive Fe$^{III}$Cit solution was ensured to always be stored in the dark and freshly prepared shortly before an experiment.

**Table 2.** Outline of all experiments with the assessed environmental conditions (each number stands for one specific condition). An experiment encompassed multiple conditions, exemplified in Figure 4. The chosen parameters were different aerosol types with different mole ratios, carrier gas, and relative humidity (RH). For each condition, there were intervals of particle irradiation and intervals without light exposure.

| Number | Aerosol type | M$_r$ | Carrier gas | RH (%) |
|:---:|:---:|:---:|:---:|:---:|
| 1 | CA | 1 | Air | 25±10 |
| 2 | CA | 1 | Air | 75±10 |
| 3 | CA | 1 | N$_2$ | 25±10 |
| 4 | CA | 1 | N$_2$ | 75±10 |
| 5 | Cu$^{II}$HCit:CA | 1:100 | Air | 25±10 |
| 6 | Cu$^{II}$HCit:CA | 1:100 | Air | 75±10 |
| 7 | Fe$^{III}$Cit:CA | 1:10 | Air | 25±10 |
| 8 | Fe$^{III}$Cit:CA | 1:10 | Air | 75±10 |
| 9 | Fe$^{III}$Cit:CA | 1:10 | N$_2$ | 25±10 |
| 10 | Fe$^{III}$Cit:CA | 1:10 | N$_2$ | 75±10 |
| 11 | Fe$^{III}$Cit:Cu$^{II}$HCit:CA | 1:0.1:10 | Air | 25±10 |
| 12 | Fe$^{III}$Cit:Cu$^{II}$HCit:CA | 1:0.1:10 | Air | 75±10 |
| 13 | Fe$^{III}$Cit:Cu$^{II}$HCit:CA | 1:0.1:10 | N$_2$ | 25±10 |
| 14 | Fe$^{III}$Cit:Cu$^{II}$HCit:CA | 1:0.1:10 | N$_2$ | 75±10 |

## 2.5   Experimental procedure and data acquisition

Table 2 summarizes the experiments conducted in this study. Experiments 1 and 2 were designed to assess background ROS levels without the presence of the chromophore Fe$^{III}$Cit, which triggers photochemical reactions in the samples. Similarly,

experiments 3 and 4 aimed to confirm that Cu$^{II}$HCit did not autonomously produce ROS, indicating that it does not act as a chromophore like Fe$^{III}$Cit. Two RH conditions (25% and 75%) and two mole ratios of Fe$^{III}$Cit:CA (1:10 and 1:100) were selected to evaluate the impact of microphysical properties such as the aerosol phase state and explore the dependencies related to the metal-to-ligand ratio. The mole ratios were chosen in order to replicate Fe$^{III}$Cit/CA experiments as done in Alpert et al. (2021), and to test two atmospherically relevant copper concentrations (approximately a tenth of the iron concentration

(Schroeder et al. (1987); Wei et al. (2019).





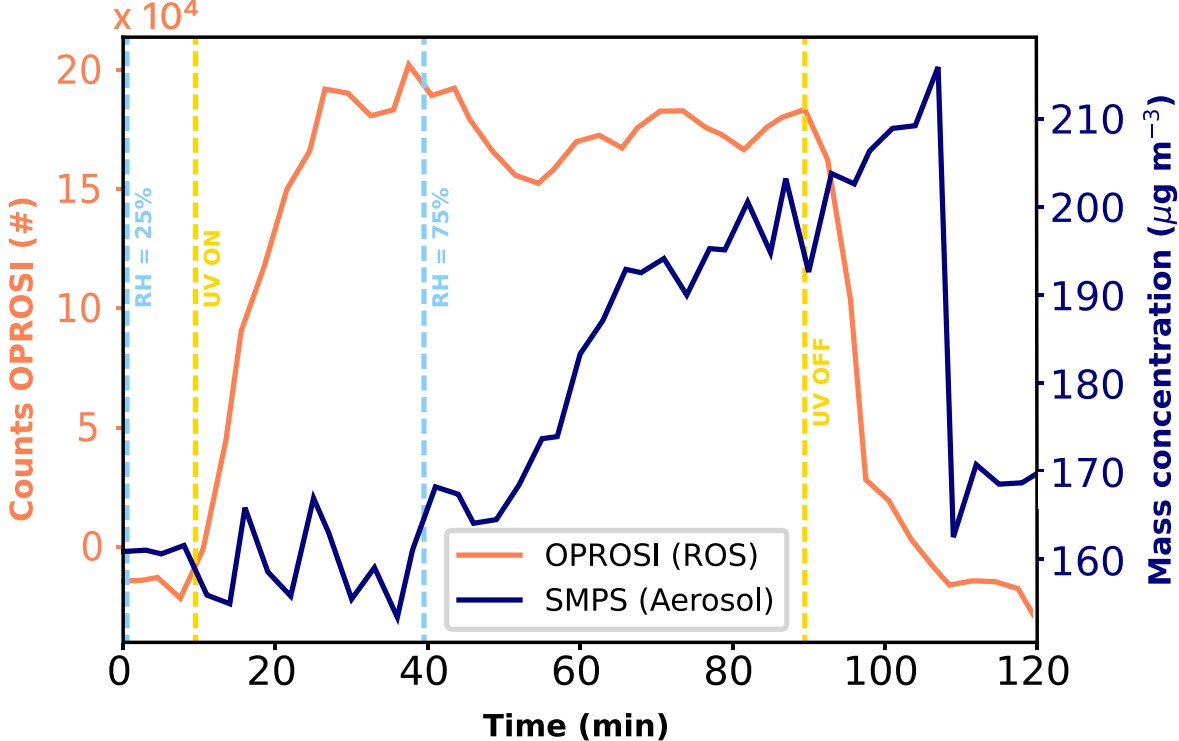

**Figure 4.** Procedure of an experiment with the OPROSI counts (coral, left y-axis) and SMPS mass concentration (navy, right y-axis) data as a function of time. The vertical dashed lines represent changes in the experimental conditions, i.e., RH (light blue) and UV radiation (yellow). The data represents an exemplary period of experiments 5 and 6 in Table 2.

To analyze the experiments, we chose distinct time periods marked by different environmental conditions such as RH and calculated the average values during those periods. In this way, the different experimental conditions described in Table 2 could be run in one sequence and qualitatively compared to each other later on. In a first step, the raw fluorescence data was blank subtracted and converted from fluorescence units ($N_{\mathrm{counts}}$) as shown in Figure 4 (coral) to ROS concentration units ($\mathrm{ROS_{DCFH}}$; nM $H_2O_2$ eq. $L^{-1}$ air). A blank measurement was performed before, during and after each experiment resulting in a second-order polynomial fit as $H_2O_2$ calibration curve with the intercept $\beta_0$ and slope $\beta_1$. Hence, the calibration curve was used to calculate the ROS concentration ($\mathrm{ROS_{DCFH}}$) as depicted in Equation 4:

$$\mathrm{ROS_{DCFH}}(\mathrm{nM\ H_2O_2\ eq.\ L^{-1}air}) = \frac{N_{\mathrm{counts}} - \beta_0}{\beta_1} \tag{4}$$

The mass-normalized ROS concentrations ($C_{\mathrm{norm}}$) are calculated following Equation 5. $\mathrm{ROS_{DCFH}}$ were normalized to the mass concentration measured with the SMPS system ($\mathrm{ROS_{DCFH}}$, see Figure 4 navy). Liquid flow rate ($F_{\mathrm{l}}$ in L $\min^{-1}$) and gas flow rate ($F_{\mathrm{g}}$ in $m^3$ $\min^{-1}$).

$$C_{\mathrm{norm}}(\mathrm{nM\ H_2O_2\ eq.\ \mu\ g^{-1}}) = \frac{\mathrm{ROS_{DCFH}} \cdot F_{\mathrm{l}}}{C_{\mathrm{aerosol}} \cdot F_{\mathrm{g}}} \tag{5}$$

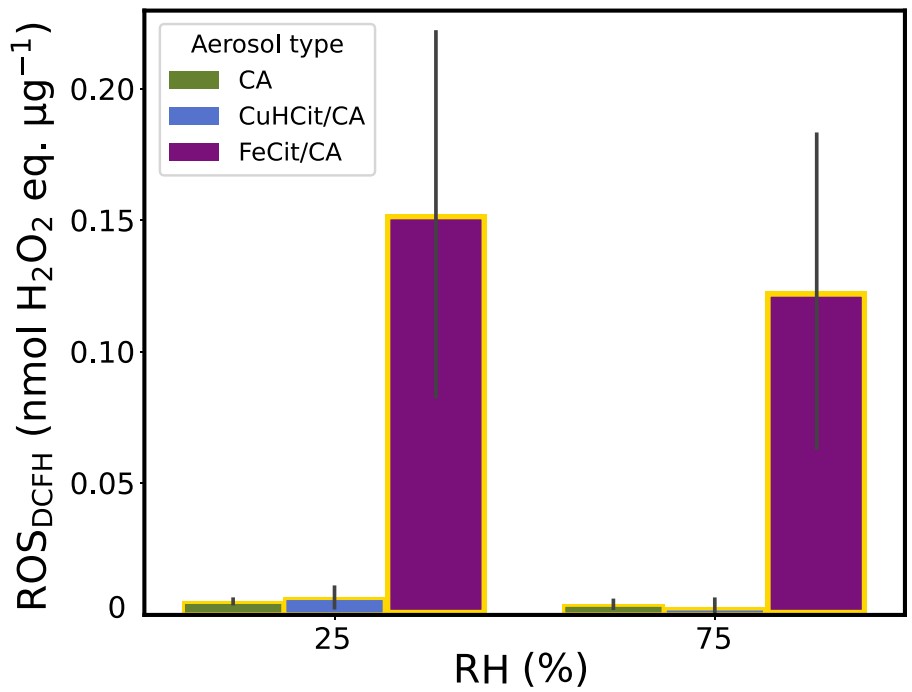

As apparent from the OPROSI data in Figure 4, the transition time in this study is a bit longer than described in Wragg et al. (2016) as we also need to account for the residence time in the aerosol flowtube (see Figure 3). Hence, in this study, the

transition time was defined as $20\,\mathrm{min}$, which also equals the OPROSI time resolution. Hence, the mass normalized OPROSI counts for each period of conditions were calculated as averages along with their standard deviations, incorporating a delay of $\approx 20\,\mathrm{min}$ after each change in conditions.

## 3   Results and Discussion

### 3.1   High ROS concentrations in UV-aged $Fe^{III}Cit/CA$ particles

**Figure 5.** ROS concentrations ($ROS_{DCFH}$) of UV-aging experiments (yellow borders) with pure CA (green), $Cu^{II}HCit/CA$ (blue) and $Fe^{III}Cit/CA$ (violet) aerosol types (experiments 1-2 and 5-8 listed in Table 2) at two different humidities (RH = 25 and 75%) in air. The gray bar denotes the standard deviation of the experiments under the same conditions. Note that the $Fe^{III}Cit/CA$ experiments were conducted in two different measurement campaigns, which might have led to the higher standard deviations.

Figure 5 introduces ROS concentrations ($ROS_{DCFH}$) of the UV-aging experiments (yellow borders) with pure CA (green), $Cu^{II}HCit/CA$ (blue) and $Fe^{III}Cit/CA$ (violet) particles under two different humidities (RH = 25 and 75%) in air used as carrier gas in the AFT. The results reveal that there was hardly any particle-bound ROS production in both reference experiments with CA and $Cu^{II}HCit/CA$ ($C_{ROS} = 2 \times 10^{-3}$-$1 \times 10^{-2}$ nmol $H_2O_2$ eq $\mu g^{-1}$) at both humidities without $Fe^{III}Cit$ as a chromophore.



The Fe$^{III}$Cit/CA experiments show substantial ROS concentrations during both RH experiments in air. At RH = 25%, ROS$_{DCFH}$
reached $\geq 0.15 \pm 0.05$ nmol H$_2$O$_2$ eq $\mu$g$^{-1}$, which is more than ten times higher compared to CA reference measurements.
The concentration at RH = 75% was $\simeq 0.03 \pm 0.05$ nmol H$_2$O$_2$ eq $\mu$g$^{-1}$ lower compared to the low RH experiment, which is
also apparent for the for the two control cases, but on a much lower level. The influence of RH on the ROS production will be
discussed below in Figure 6.

The findings verified that ROS production in CA particles as a SOA proxy is dominated by the photochemistry initiated
by the photolysis of the iron citrate complex. This aligns with the high UV absorbance spectrum at wavelengthes around $\lambda =$
365 nm for Fe$^{III}$Cit (Seraghni et al., 2012), and the absence of measurable absorbance for copper-organic complexes at $\lambda \geq$
250 nm (Seraghni et al., 2021) and of CA (Seraghni et al., 2012). To put the values in context, these ROS$_{DCFH}$ concentrations
also remained in the range of measurements carried out similarly. Research by Campbell et al. (2023) examined both online
and offline measurements of the oxidative potential of SOA, where the OPROSI data of SOA with $\beta$-pinene and naphthalene
180 as precursors indicated ROS$_{DCFH}$ = 0.1-0.25 nmol H$_2$O$_2$ eq $\mu$g$^{-1}$. Also, with the OPROSI, Campbell et al. (2021) measured
winter and summer 24 h averaged ROS$_{DCFH}$ from PM$_{2.5}$ filter samples in Beijing, China. The ROS$_{DCFH}$ ranged from $3 \times 10^{-3}$-
$1 \times 10^{-2}$ nmol H$_2$O$_2$ eq $\mu$g$^{-1}$. Other ambient particle-bound ROS concentrations were quantified by Arangio et al. (2016) with
electron paramagnetic resonance spectra corresponding to similar concentrations. Inscribed in the literature, we can confirm
the usability of mimicking photochemical aging in the AFT setup toward ROS measurements. Although, the primary objective
185 of this study was to qualitatively assess the influence of copper, humidity and oxygen supply on the build-up of ROS in iron
containing SOA, we first contrast experiments of dark and UV aged Fe$^{III}$Cit/CA particles under both humidity scenarios (RH
= 25 and 75%) and two different carrier gas conditions.

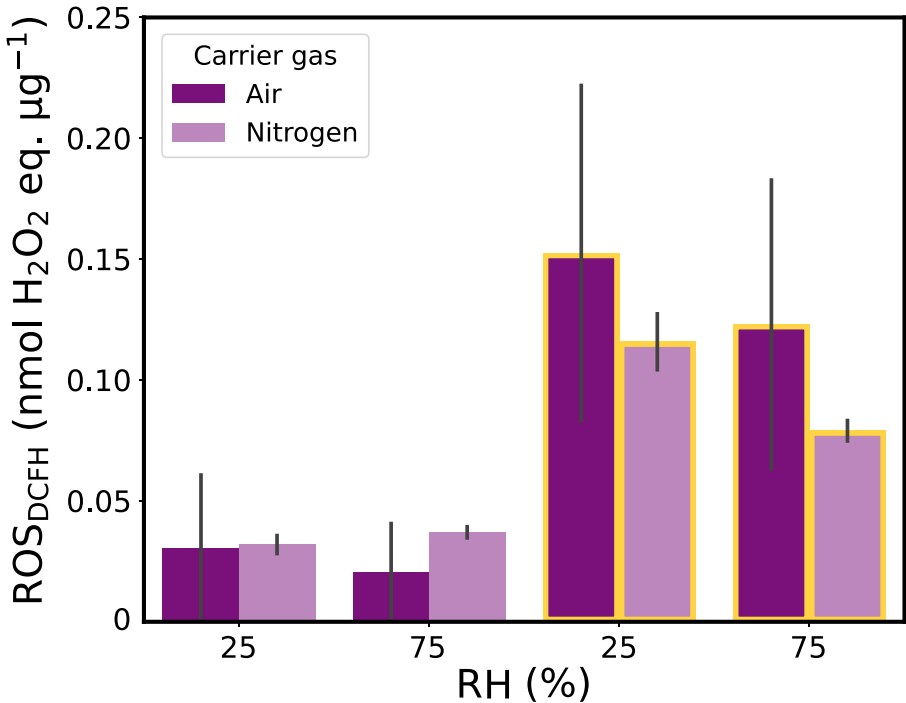

**Figure 6.** ROS concentrations ($ROS_{DCFH}$) of dark and UV-aging (yellow borders) experiments with $Fe^{III}Cit/CA$ proxies (experiments 7-11 listed in Table 2) at two different relative humidities (25 and 75% RH) in air (violet) and nitrogen (light violet). The gray bar denotes the standard deviation of the experiments under the same conditions.

The UV-aged samples in air (violet) and in nitrogen (light violet) are depicted with yellow borders under RH = 25 and 75% in Figure 6. Regardless of RH conditions and the type of carrier gas, $ROS_{DCFH}$ in particles exposed to UV light were approximately ten times higher than those in particles that were not exposed to any light. The highest concentrations were measured under 25% RH in air ($ROS_{DCFH} = 0.15 \pm 0.05\,\mathrm{nmol\ H_2O_2\ eq}\ \mu g^{-1}$, as already depicted in Figure 5). Particles photochemically aged under low humidity (25% RH) showed higher $ROS_{DCFH}$ than those aged at 75% RH with the same carrier gas, which is consistent with the effect of RH on the data for the three aerosol types shown in Figure 5 above.

At low humidity, CA particles become highly viscous (Kasparoglu et al., 2022; Reid et al., 2018), and thus diffusion of, e.g., oxygen from the gas to the particle phase becomes limited. Alpert et al. (2021) found that the photochemical aging of such viscous particles produces a large amount of carbon-centered free radicals (CCFR, see Figure 1) in their interior. These CCFRs are a sink for oxygen to generate ROS, mostly through a first generation of peroxy radicals. If this oxygen demand exceeds the oxygen supply by uptake from the gas phase (driven by the rather low solubility of oxygen) and diffusion from the surface towards the interior (reduced by low diffusivity in the viscous SOA medium at low RH), anoxic conditions are induced in the interior of the particles (Alpert et al., 2021). By modeling this system, they simulated $ROS_{DCFH} \simeq 8 \times 10^{-3}\,\mathrm{nmol\ H_2O_2\ eq}\ \mu g^{-1}$ for 20% RH. However, our online measurements also showed high $ROS_{DCFH}$ in particles aged in





nitrogen in which we do not assume oxygen being present in the bulk. We argue that a limited amount of oxygen in the particle phase (from the nebulized solution, diffusion through the permeable tubing and/or the $N_2$ gas flow not being 100% pure) was sufficient to oxidize the CCFR (see R2 in Table 1) and start ROS cycling reactions. Thus, we assume a reduction of the $O_2$ concentration by a factor of 100 in the $N_2$ carrier gas experiments in comparison to air. In addition, we cannot rule out that $ROS_{DCFH}$ also includes the detection of CCFRs by a HRP reaction with CCFR. If this is the case, there is a higher fraction of CCFRs among all ROS under anoxic conditions, which we were not able do discriminate with the OPROSI.

Even in humid conditions, $ROS_{DCFH}$ levels remained high, albeit lower than in dry conditions, a fact we aim to interpret next. The $ROS_{DCFH}$ concentrations were more than two orders of magnitude higher than the ones simulated by Alpert et al. (2021) ($ROS_{DCFH} \simeq 3 \times 10^{-4}$ nmol $H_2O_2$ eq $\mu g^{-1}$). On the one hand, Alpert et al. (2021) only considered $H_2O_2$ formed from the self-reaction of $HO_2$ that is eliminated from the first generation peroxy radical in the $\alpha$ position to the alcohol group. All other peroxy radical sources from secondary OH chemistry were not considered. In addition, the $H_2O_2$ levels quoted above were steady state concentrations simulated while the particles were still exposed to UV light and thus, ROS was continuously consumed by $Fe^{2+}$, which was not the case in our experiments. In air experiments, during the dark flow period ($\approx$ 1-2 s) downstream of the AFT and before mixing with HRP in the OPROSI particle collector, oxygen reacts with remaining CCFRs to generate ROS. From a physicochemical perspective, it is expected that within the low viscosity particles, there would be higher diffusion rates for oxygen (to diffuse in) and the more volatile members of ROS. This implies a high level of oxygen within the bulk phase and a rapid exchange of ROS with the gas phase. The results suggest that these processes are somewhat balanced by increased diffusional loss of ROS and/or more rapid reoxidation of Fe(II) induced by ROS. If this were not due to experimental limitations, it would be significant, as it suggests ROS production in both liquid and viscous SOA particles through the pathways shown in Figure 1. This would also contradict the findings of Alpert et al. (2021).

The $ROS_{DCFH}$ levels in non-aged $Fe^{III}Cit/CA$ particles under $N_2$ conditions were slightly higher compared to air (lower standard deviation). This could be due to impurities, artifacts, or the inherent dark CCFR production that did not oxidize in $N_2$ conditions. In summary, photochemically aged $Fe^{III}Cit/CA$ particles produced a considerable amount of particle-bound ROS. The different carrier gases and RH conditions only slightly changed the $ROS_{DCFH}$ levels, although low humidity (25% RH) and the presence of oxygen in the environment led to higher concentrations.

### 3.2 Oxygen limitation reduced ROS concentration in copper containing particles

After evaluating the UV-aged and non aged $Fe^{III}Cit/CA$ particles, we now turn our attention to the results of UV-aged particles that also contain copper. Figure 7 includes $ROS_{DCFH}$ data from $Fe^{III}Cit/Cu^{II}HCit/CA$ (blue) UV-aging experiments, alongside the $Fe^{III}Cit/CA$ (violet) presented above in Figure 6. The trend towards lower $ROS_{DCFH}$ at high humidity (RH = 75%) was also observed for UV-aged $Fe^{III}Cit/Cu^{II}HCit/CA$ particles. Using air as carrier gas, the $ROS_{DCFH}$ levels in $Fe^{III}Cit/Cu^{II}HCit/CA$ particles were about $0.05$ nmol $H_2O_2$ eq $\mu g^{-1}$ lower compared to $Fe^{III}Cit/CA$, but similar (within standard deviations) to the $Fe^{III}Cit/CA$ particles UV-aged in $N_2$. Furthermore, $ROS_{DCFH}$ for copper-containing samples was markedly lower under both humidity conditions when measured in $N_2$ compared to all other UV-aged samples. The gradual decrease of $ROS_{DCFH}$ from UV-aged $Fe^{III}Cit/CA$ (air) to $Fe^{III}Cit/Cu^{II}HCit/CA$ particles ($N_2$) was observed in both humidity regimes. At 25% RH, the





particles will have higher viscosity, which may lead to an accumulation of CCFR in bulk due to very limited oxygen diffusion from the gas phase, before reacting to ROS during dissolution in OPROSI, which is consistent with Alpert et al. (2021). Hence, the availability of oygen from the gas phase is not influencing the ROS production at such low RH. Instead, at higher humidity (RH = 75%), $ROS_{DCFH}$ shows an increased variability from air to $N_2$ as carrier gas, probably due to the liquid particles,

allowing for faster diffusion of oxygen and loss of ROS to the gas phase. Next, we explore how copper might impact ROS levels in photochemically aged $Fe^{III}Cit/CA$ particles.

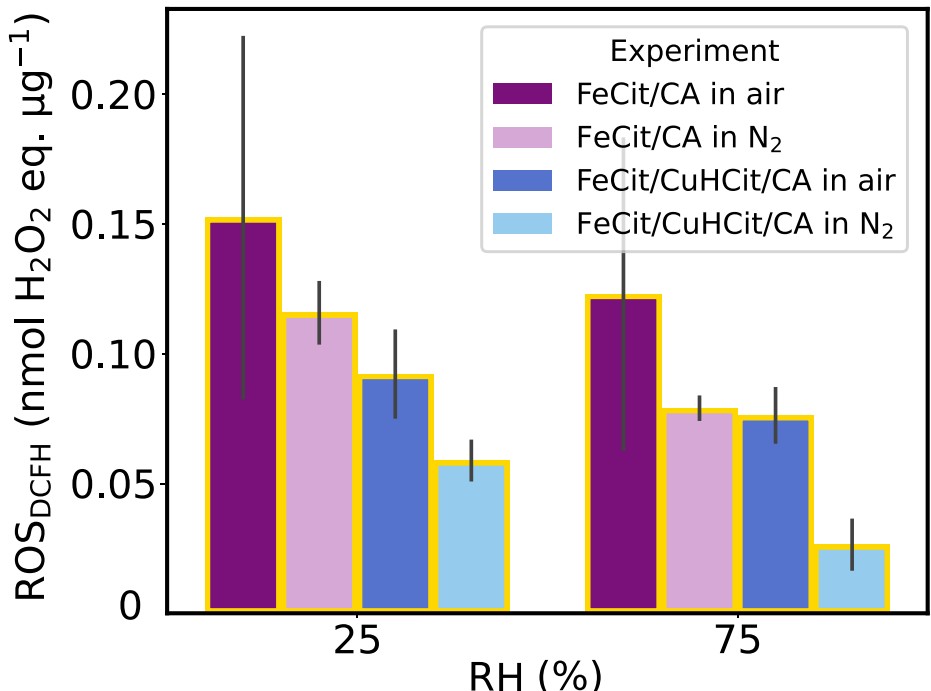

**Figure 7.** ROS concentrations ($ROS_{DCFH}$) from $Fe^{III}Cit/CA$ (violet, light violet) and $Fe^{III}Cit/ Cu^{III}HCit/CA$ (blue, light blue) UV aging experiments (yellow borders) at changing experimental conditions (RH and carrier gas). The gray bar denotes the standard deviation of the experiments under the same conditions. Both experiments were carried out in a campaign in 2023.

The findings back previous efforts to model $Fe^{III}$ reoxidation in photochemically aged $Fe^{III}Cit/ Cu^{II}HCit/CA$ particles. A lower $Fe^{III}Cit$ quantum yield (R1 in Table 1) and/or a copper-triggered ROS reduction mechanism could explain the data (Kilchhofer et al., 2024). Other research groups also identified copper-induced ROS reduction mechanisms in an aerosol system

(Ervens et al., 2003; Mao et al., 2013; Shen et al., 2021). Mao et al. (2013) for instance proposed that Cu-Fe redox coupling in aqueous aerosols induced radical loss. In this case, Cu catalyzed $HO_2$ to $H_2O_2$ at low pH and $H_2O_2$ then oxidized $Fe^{2+}$ resulting in a net ROS loss. This can also be followed by a summarized chemical mechanism that includes faster iron(III) reoxidation, which could explain faster ROS depletion in the presence of copper (Kilchhofer et al., 2024). In more detail, $Cu^{2+}$




could consume ROS via $Cu^{2+} + HO_2^{\bullet} \longrightarrow Cu^+ + O_2 + H^+$ and $Cu^{2+} + O_2^{\bullet-} \longrightarrow Cu^+ + O_2$. These arguments are further

supported by non aging experiments.

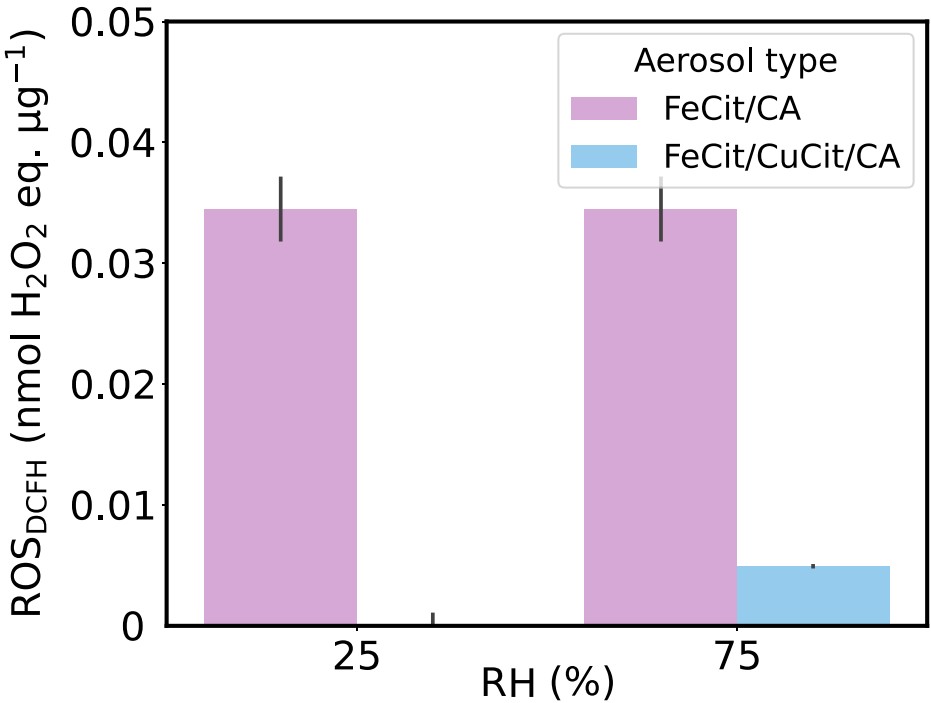

**Figure 8.** ROS concentrations ($ROS_{DCFH}$) of non-aged $Fe^{III}Cit/CA$ (light violet, same as shown in Figure 6) and $Fe^{III}Cit/ Cu^{II}HCit/CA$ particles (light blue) in nitrogen at different humidities (RH). The gray bar denotes the standard deviation of the experiments at same conditions.

Figure 8 resumes the non-aged $Fe^{III}Cit/CA$ ROS concentrations shown in Figure 5 (light violet) including the non-aged $Fe^{III}Cit/Cu^{II}HCit/CA$ concentrations in nitrogen (light blue). It seems that the copper-induced ROS oxidations consumed all of the remaining ROS in the bulk of the copper-containing particles, as argued before, in UV-aged particles.

## 4    Conclusion

Reactive oxygen species (ROS) present in aerosol particles correlate with their toxicity. Here, we study the potential production of ROS by photochemically induced redox chemistry in iron/copper citrate particles. We experimentally mimicked such a process with an aerosol flow tube (AFT) in which UV-aging was probed and different atmospheric conditions were tested. An online particle-bound ROS instrument (OPROSI) was used to assess ROS production during the different UV and dark periods investigating various aerosol types.





In conclusion, this study showed that photochemically aged $Fe^{III}Cit/CA$ particles generate significant levels of ROS, with production largely driven by the photolysis of $Fe^{III}Cit$. We found that UV-aged CA aerosol containing 10 mole % $Fe^{III}Cit$ generated ROS concentrations on the order of $0.1 \, nmol \, H_2O_2 \, eq \, \mu g^{-1}$. The experiments demonstrated that ROS concentrations were highest at low relative humidity (RH = 25%), in the presence of air and pure $N_2$ as a carrier gas (within the standard deviation). At such RH, the particles become higher viscous, which may lead to an accumulation of carbon centered free rad-

icals (CCFRs) in bulk due to very limited oxygen diffusion from the gas phase, before reacting to ROS during dissolution in OPROSI, which is consistent with Alpert et al. (2021). Hence, the availability of oxygen from the gas phase is not influencing ROS production at such a low RH. Instead, at higher RH (75%), $ROS_{DCFH}$ shows an increased variability from air to $N_2$ as carrier gas and an overall decreased production compared to 25% RH, probably due to the reduced viscosity of the CA particles, allowing for faster diffusion of oxygen and loss of ROS to the gas phase.

The role of copper was investigated in $Fe^{III}Cit/Cu^{II}HCit/CA$ particles, with the results showing that these copper-containing particles produced lower ROS concentrations than iron-only particles. This suggests copper involvement in ROS depletion mechanisms, such as Cu-Fe redox coupling, which accelerates the consumption of ROS. Furthermore, the findings align with the findings by Kilchhofer et al. (2024) in terms of lower ROS production induced by a lower iron(II) quantum yield ($\phi$) and Cu-induced ROS oxidation reactions, which reduces ROS concentrations in the particle phase.

Overall, the study highlights the complex interplay of humidity, oxygen availability, and metal catalysis to control ROS production in secondary organic aerosols. These insights contribute to a deeper understanding of atmospheric aging processes and the factors influencing aerosol OP, with implications for air quality and human health. We are aware of the limitations by too much focus on the peroxides by only measuring with DCFH as an acellular assay and thus, recommend using other suitable assays in detecting the whole range of particle-bound ROS.



**Appendix A: Additional Figure**

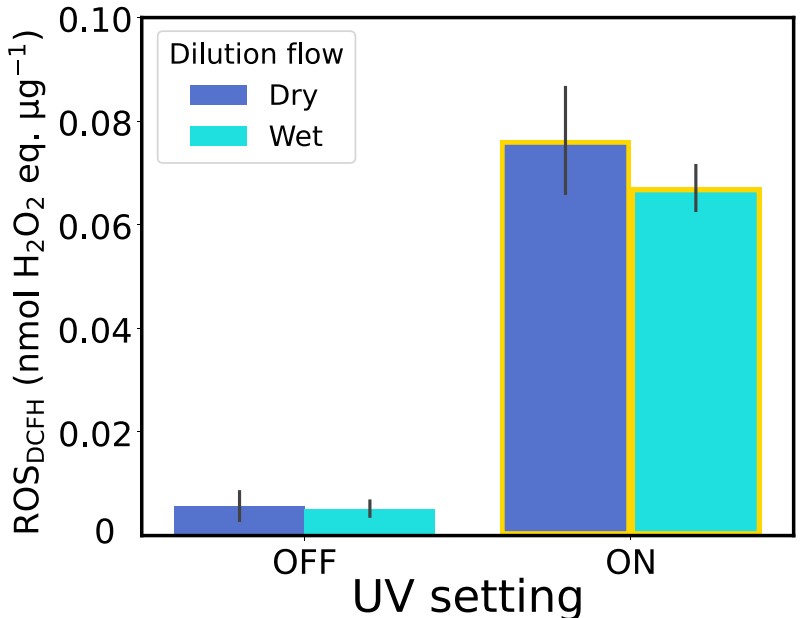

**Figure A1.** ROS concentrations (ROS$_{DCFH}$) from Fe$^{III}$Cit/Cu$^{II}$HCit/CA experiments in air at different humidities (RH) and dilution flow conditions (dry = blue, wet = cyan). The gray bar denotes the standard deviation of the experiments at same conditions.

Considering the experimental setup described in Figure 2, one could ask if RH was present after adding a dry dilution flow. Therefore, we aimed to rule out that a dry dilution gas flow could affect ROS$_{DCFH}$ measurements. With the results presented in Figure A1 we can rule out such an influence, as no variation was detected in the measured ROS$_{DCFH}$ when using a dry or wet dilution flow in an experiment carried out at RH = 75%.

*Author contributions.* MA, MK and KK designed the research. AB, BU and KK carried out the experiments. AB, BU and KK analyzed the data. KK conducted the data processing and wrote the manuscript with significant inputs for the Methods by AB and BU.

*Competing interests.* The authors declare that they have no known competing financial interests or personal relationships that could have appeared to influence the work reported in this paper.

*Acknowledgements.* The authors thank the Swiss National Science Foundation for financial support with grant numbers 188662 and 192192.





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
