# Peer review of "Reactive Oxygen Species Build-up in Photochemically Aged Iron-and Copper-doped Secondary Organic Aerosol Proxy"

_Aerosol Research, 2024_

## Author Response (AR1)

**RC1:**

The manuscript, "Reactive Oxygen Species Build-up in Photochemically Aged

Iron-and Copper-doped Secondary Organic Aerosol Proxy" by Kilchhofer et al., conducted a study to measure photochemically induced reactive oxygen species (ROS) production on SOA proxy with metal complexes.

The manuscript overall is well-written and easy to follow. However, the manuscript could be improved by providing a more thorough explanation of the methods and results, as well as addressing a few limitations before it is ready for final publication.

Thanks for the opportunity to review this interesting manuscript.

**Major comments:**

1. The authors provide a clear explanation for their choice of Cu and Fe particles to evaluate ROS production in the atmosphere. However, there is no justification for selecting citric acid (CA) as a surrogate for SOA. Can CA effectively represent SOA? The authors should address this point in the introduction.

We acknowledge this comment and regret that this was not outlined clearly. Hence, we added the following text to the introduction and changed the sentence on line 44 with ' '.

Indirect measurements and model results reported ROS build-up of metal complexed 'citric acid (CA)' during photochemical aging processes (Dou et al., 2021, Alpert et al., 2021, Kilchhofer et al., 2024). CA was and will be used here as SOA proxy, because the chemical composition of SOA is very diverse and highly complex and thus, it is impossible to fundamentally describe individual chemical processes in SOA material. We will add the following text in addition:

'CA comprises of three carboxylic acid and one tertiary alcohol functional group, which is typical for SOA. CA has also been directly identified in aerosol particles (Graham et al., 2002, Decesari et al., 2002, Boreddy et al., 2022). Because CA has well defined microphysical properties and does not easily crystallize at low relative humidity, it has been frequently used as model substance for atmospheric chemistry experiments (Murray et al., 2010, Dou et al., 2021, Alpert et al., 2021, Kilchhofer et al., 2024). Heterogeneous photochemistry initiated by photolysis of iron carboxylate complexes contributes to the oxidant budget in atmospheric particles and thus leads to the formation of particle-bound ROS (Corral-Arroyo et al., 2018).'

2. As mentioned by the authors in the introduction, the rationale behind using the DCFH assay to measure ROS in SOA particles is explained. Since the DCFH assay measures only specific ROS in SOA particles, the authors should provide more details on the potential limitations of OPROSI in measuring ROS from aerosols that contain multiple types of ROS.

We like to emphasise that there is no analytical method that can quantify all possible oxidising components in organic aerosols, mainly due to the complex compositions of SOA with thousands of often highly oxidised components, most of which have unknown structures. A quantification of ROS via DCFH and horseradish peroxidase (HRP) is mainly sensitive to peroxides, hydroperoxides including $H_2O_2$ and peroxyacids and possibly other short-lived ROS such as radicals. To clarify this aspect, we added the following sentence on line 62: 'DCFH is sensitive to $H_2O_2$ and organic peroxides (Fuller et al., 2014), but not to redox-active transition metals like iron and copper (Campbell et al., 2023.The sensitivity of the DCFH assay towards radicals is unclear.'

In addition, we were specifically interested in the photochemical mechanism occurring in the iron complexed CA particles. Hence, we were interested in the reactions shown in Table 1 with products such as $H_2O_2$ that react efficiently with DCFH. Furthermore, we could not use ascorbic acid (AA) as an assay, because copper would have intrinsically reacted with AA leading to a signal that would dominate of that by the particle-bound ROS.

**Minor comments:**

3. The authors define the ROS concentration ($ROS_{DCFH}$) unit as $nM\ H_2O_2\ L^{-1}\ air$ and the mass-normalized ROS concentration ($C_{norm}$) unit as $nM\ H_2O_2\ eq.\ \mu g^{-1}$. However, in the results and discussion sections, as well as in some figures, these two abbreviations are used interchangeably. For instance, in Figure 5, the y-axis unit is labeled as $ROS_{DCFH}$ ($nmol\ H_2O_2\ eq.\ \mu g^{-1}$). According to the authors' definition, the unit for $ROS_{DCFH}$ should be volume-normalized ($nM\ H_2O_2\ eq.\ L^{-1}\ air$). Additionally, $nmol$ and $nM$ are distinct units. Please review and clarify this inconsistency.

Thank you for spotting this. We corrected the unit mistake in the definitions on line 159-163 to $ROS_{DCFH}$(nmol $H_2O_2$ eq.) and to $C_{norm}$(nmol $H_2O_2$ eq. $\mu g^{-1}$). Additionally, we changed the y-axis of the mass-normalized data in all the Figures to $C_{norm.}$

4. Line 231 to 233: Figure 7 indicate about ROSDCFH level of FeCit/CA and FeCit/CuHCit/CA under air and $N_2$ conditions, with 25% and 75% RH. The authors explain "$ROS_{DCFH}$ in FeCit/CuHCit/CA were about 0.05 nmol $H_2O_2$eq $ug^{-1}$ lower compared to FeCit/CA.". However, the observed difference in

ROS$_{DCFH}$ level between FeCit/CA and FeCit/CuHCit/CA under air conditions appears to be smaller than the stated 0.05 nmol H$_2$O$_2$*eq ug*$^{-1}$. Please review this discrepancy and clarify.

Here, we do not see your observed discrepancy. However, we agree that this sentence is not easy to follow and thus, we clarified the sentences in this context on line 236: 'Using air as carrier gas, the ROS$_{DCFH}$ levels in FeCit/CuHCit/CA particles were about 0.05 nmol H$_2$O$_2$*eq ug*$^{-1}$ lower compared to FeCit/CA. However, the ROS concentration is on the same level (within standard deviations) compared to the FeCit/CA particles UV-aged in N$_2$.'

5. Line 242: "The findings back previous efforts to model FeIII reoxidation in photochemically aged~." can be reworded into "These findings support previous efforts to ~."

Thank you and we tried to make it even more clear. We changed the sentence on line 246 to: 'The findings in copper-containing particles support previous efforts to ... .'

**RC2**:

This study investigated the reactive oxygen species (ROS) produced through photochemical aging of SOA proxy (citric acid) with metals and further examines key factors influencing ROS generation, including RH, oxygen availability, and the presence of copper. The manuscript is well-written and logically sound. However, I have several concerns that need to be addressed before it can be considered for publication.

1. One main limitation is that the manuscript does not clearly articulate the study's novelty. What new scientific insights does this work provide? At present, it appears to be an experimental validation of previous simulation studies.

We thank for pointing out that the manuscript does not sufficiently express the novelty. We like to stress here on the fact that we, for the first time, measured particle-bound ROS concentrations of a photochemically aged SOA proxy, which proves the findings by previous simulation studies. Hence, we could show that modeling studies simulated valid particle-bound ROS levels, which was not proven previously by experimental work. Additionally, our goal was to find another observable helping us to elaborate the complex photochemical mechanism of iron-complexes CA particles and find the influence of copper on top of that. This was recently studied by other modeling and experimental work by our and other groups. It will also help to better model particle-bound ROS concentration in other SOA proxies and in bigger atmospheric scale chemistry models.

2. The experimental design is too simple. For example, only one molar ratio of FeIIICit:CA, CuIIHCit:CA, FeIIICit:CuIIHCit:CA was investigated. While the selected ratio is reasonable and mimics real atmospheric conditions, the actual atmospheric FeIIICit:CA ratio likely varies spatially. The authors should assess the effect of varying this ratio on ROS production to enhance the study's significance.

   There are many parameters that could potentially affect the reactivity of metal-organic aerosol components and the formation of ROS, such as type of metal, mixtures of metals, type of organic compound, mixtures of organics, metal-organic ratio, relative humidity, oxidation scheme, carrier gas, etc.. We explore a range of these parameters and their effects on ROS formation (see Figure 5 – 8) but it would be beyond the scope of this work to cover this parameter space in its entirety. Nonetheless, we agree that the FeCit:CA ratio is an important parameter for the ROS formation investigated here and we now add an additional figure in the Appendix illustrating that the ROS formation decreases by a factor of 3-4 when the FeCit:CA ratio decreases from 1:10 to 1:100.

3. The rationale for using citric acid as an SOA proxy is not clearly justified. Is citric acid an oxidation product of any volatile organic compounds in the atmosphere? How abundant is it? The authors should clarify these points.

   Thank you for this comment. Please refer to our answer to RC 1 above regarding the same point.

Other comments:

4. Line 18: Please clarify why especially India and China. Is this due to their high PM pollution levels or large populations?

   Ok, we see what you mean, and we deleted the parenthesis.

5. Line 28: The statement regarding oxidative potential (OP) should be revised. OP was not first defined by Bate et al. (2019).

   Thank you for picking this up. We have revised this and corrected it with a general knowledge term accordingly on line 27: 'This process is defined as the capability of PM to induce oxidative stress.'

6. Here and there (e.g., the captions of Figures 1 and 2), you wrote $Cu^{III}HCit$, please correct to $Cu^{II}HCit$.

   Thank you for spotting this. We corrected it as suggested.

7. Lines 138-140, according to Table 2, Experiments 1 to 4 are background ROS tests? And Experiments 5 and 6 designed to confirm that CuIIHCit did not autonomously produce ROS?

   This is correct. We revised the numbering in Table 2 and in the text, accordingly. We also added the FeCit:CA 1:100 experiments as shown in the appendix with experiments number 7 and 8.

8. Line 141, it states that two mole ratios of FeIIICit:CA (1:10 and 1:100) were selected, but only one ratio is present in Table 2. Also, the reported ROS results are derived from only one ratio. This inconsistency should be addressed.

   Good catch, thank you. We now made the link to the FeCit:CA 1:100 experiments shown in the appendix and corrected a typo in Table 2 (Number 7 and 8 to FeCit:CA 1:100 instead of CuCit:CA 1:100 (number 5 and 6 now).

9. Line 179, replace "oxidative potential" to ROSDCFH for consistency.

   We changed 'oxidative potential' to ROS levels on line 185.

10. Line 222, the authors should perform a statistical analysis (e.g., t-test) and provide p values to quantify the significance of differences observed under various experimental conditions.

    Ok, we acknowledge your proposition of performing a statistical analysis here. However, we expressed the significance of the results for various experimental conditions by showing the standard deviation of each experiment. While the errors based on the measurement precision are in principle fairly small, the overall error estimated, is dominated by systematic errors from e.g., analysis of the aerosol mass concentration (around 20%), or the collection efficiency of ROS and others. This was the basis for adding error bars to all results. Hence, we changed part of the text on line 227 to:
    'The ROS levels in non-aged FeCit/CA particles are by a factor of 3-5 smaller than those after photochemical aging and no clear trend is observed between $N_2$ and air carrier gas conditions (in contrast to UV-aging conditions discussed above), which might be in part due to the very low overall ROS concentrations or due to impurities, artifacts, or the inherent dark CCFR production that did not oxidize in $N_2$ conditions.'

11. Line 238: typo, change "oygen" to "oxygen"

    We corrected this typo on line 242.

12. Line 279: what assays can be recommended to detect the whole range of particle-bound ROS? The authors should discuss potential methodologies.

As mentioned above, there is no individual ROS analysis method available, which quantifies all possible ROS components in aerosol particles. DCFH is an assay which is specifically sensitive to peroxides, a compound class likely important in the reaction system investigated here and we could not use AA because of its reactivity towards transition metals as discussed as well.

---

## Author Response (AR2)

**Revision of final comments from editor:**

Thank you for the second review round and we are pleased to revise the final minor comments from your part. The revised version is stated in blue below your comments:

• Lines 62-64 "Here, we focus on examining the formation of particle-bound ROS within an iron and/or copper CA particles induced by complex photolysis under changing atmospheric conditions."

In this study, we examine the generation of particle-bound ROS in CA particles containing iron and/or copper because of photolytic processes influenced by varying atmospheric conditions.

• Figure 1 caption "... The left pathway exemplary depict an OA growth..."

On the left pathway (blue particles) an OA particle growth in high RH conditions is drawn, implying a liquid OA phase, whereas on the right the particles (green) experience low RH conditions that lead to highly viscous organic phase.

• Lines 83-85 "As schematically shown in Figure 1, we expected lower particlebound ROS concentrations in aerosols photochemically aged under humid compare to dry conditions..."

As depicted in Figure 1, we anticipated that aerosols undergoing photochemical aging in humid conditions would exhibit lower concentrations of particle-bound ROS compared to those in dry conditions. This is because effective diffusion aids in the exchange between gas and particle phases, leading to greater loss to the gas phase.

• Line 220 "The low levels of oxygen must..."

The low oxygen levels must have been sufficient to oxidize the CCFR (refer to R2 in Table \ref{Chap\_3\_Tab\_1}) and initiate ROS cycling reactions. Please define the abbreviation "AA" used in Lines 90-91.

• Please make sure the parenthesis in placed correctly in Lines 216-222 "However, our online measurements also showed high Cnorm in particles aged in nitrogen thus normally in absence of oxygen. Though we caution that a limited amount of oxygen in the particle phase (from the nebulized solution, diffusion through the permeable tubing and/or oxygen traces in the N2 gas flow not being may have led to only about a factor of:100 % decrease of oxygen when switching from air to N2). Good point. We have deleted the paranthesis and used 'for instance' instead: 'Though we caution that a limited amount of oxygen in the particle phase for instance from the nebulized solution, diffusion through the permeable tubing and/or oxygen traces in the N\_2 gas flow may have led to only about a factor of 100 decrease of oxygen when switching from air to N\_2.'